# Current Update on the Clinical Utility of MMSE and MoCA for Stroke Patients in Asia: A Systematic Review

**DOI:** 10.3390/ijerph18178962

**Published:** 2021-08-25

**Authors:** Julia Khaw, Ponnusamy Subramaniam, Noor Azah Abd Aziz, Azman Ali Raymond, Wan Asyraf Wan Zaidi, Shazli Ezzat Ghazali

**Affiliations:** 1Clinical Psychology and Behavioral Health Program, Faculty of Health Sciences, Universiti Kebangsaan Malaysia, Kuala Lumpur 50300, Malaysia; P102147@siswa.ukm.edu.my (J.K.); shazli_ezzat@ukm.edu.my (S.E.G.); 2Centre for Healthy Ageing and Wellness, Faculty of Health Sciences, Universiti Kebangsaan Malaysia, Kuala Lumpur 50300, Malaysia; 3Department of Family Medicine, Universiti Kebangsaan Malaysia Medical Centre, Bandar Tun Razak, Kuala Lumpur 56000, Malaysia; azah@ppukm.ukm.edu.my; 4Neurology Unit, Department of Internal Medicine, Universiti Teknologi MARA, Shah Alam, Selangor 40450, Malaysia; drraymondazmanali@utim.edu.my; 5Neurology Unit, Department of Medicine, Universiti Kebangsaan Malaysia Medical Centre, Bandar Tun Razak, Kuala Lumpur 56000, Malaysia; wan.asyraf.wan.zaidi@ppukm.ukm.edu.my

**Keywords:** stroke, cognitive impairments, cognitive screening, education, Asia

## Abstract

Objective: Primary care clinicians in Asia employed the Mini-Mental State Examination (MMSE) and Montreal Cognitive Assessment (MoCA) to aid dementia diagnosis post-stroke. Recent studies questioned their clinical utility in stroke settings for relying on verbal abilities and education level, as well as lack of consideration for aphasia and neglect. We aimed to review the clinical utility of the MMSE and MoCA for stroke patients in Asia and provide recommendations for clinical practice. Methods: PubMed, Scopus, Web of Science, and Science Direct were searched for relevant articles. Included studies were assessed for risk of bias. RevMan 5.4 was used for data synthesis (sensitivity and specificity) and covariates were identified. Results: Among the 48 full-text articles reviewed, 11 studies were included with 3735 total subjects; of these studies, 7 (77%) were conducted in China, 3 (27%) in Singapore, and 1 (9%) in South Korea. Both the MMSE and MoCA generally showed adequate sensitivity and specificity. Education was identified as a covariate that significantly affected detection accuracy. Due to heterogeneity in cutoff scores, methodologies, and languages, it was not feasible to suggest a single cutoff score. One additional point is recommended for MoCA for patients with <6 years of education. Conclusion: Clinicians in Asia are strongly recommended to consider the education level of stroke patients when interpreting the results of the MMSE and MoCA. Further studies in other Asian countries are needed to understand their clinical value in stroke settings.

## 1. Introduction

The risk of dementia in the first year after stroke is 50% greater than in the general population [1] and about 40% of stroke patients will present with mild cognitive impairments [2]. The MMSE (Mini-Mental State Examination) [3] and MoCA (Montreal Cognitive Assessment) [4] are the most used screening tools for cognitive impairments after stroke [5]. Both screening tests were originally designed to screen for dementia and mild cognitive impairments (MCIs). The diagnostic criteria for these conditions are based on the cognitive presentation of Alzheimer’s disease (AD), where memory deficits are prominent [6,7]. However, unlike AD, stroke patients show more salient frontal/executive deficits, e.g., attention and cognitive flexibility [8,9]. The term vascular cognitive impairments (VCIs) was proposed to represent a continuum of cognitive deficits of vascular etiology [7] including post-stroke dementia (PSD) [10], thereby delineating it from AD.

The different cognitive profiles of AD and stroke suggest that the MMSE and MoCA may not be useful in stroke settings, as they do not consider impairments intrinsic to stroke, i.e., aphasia, neglect, and apraxia [5,11,12,13,14,15]. For instance, both tests place a high load on verbal abilities, which can be problematic for aphasic patients in areas where language is required to perform well [16,17]. In contrast, stroke patients who retain their language abilities, such as those with right ischemic lesion, might give a false impression of normal cognition [18]. The MMSE seems to fare worse than the MoCA in detecting post-stroke cognitive impairments (PSCIs) due to its reliance on language [19]. For example, performance on calculation and attention in the MMSE varies across Asian countries [20], possibly because some languages have a higher phonological load for number processing [21].

In addition to the inherent limitations of the MMSE and MoCA, it is also critical to select a valid cutoff score for PSCI due to its influence on detection accuracy. Many studies have found the cutoff of 26 in the MoCA [4] to be inadequate in addressing cognitive impairments in stroke settings. Rather, optimal values were shown to range from 19 to 27, conditional on whether screening was conducted in the acute or chronic phase of stroke [22,23]. Preliminary evidence in Asia suggests that the MoCA is more sensitive than the MMSE in predicting cognitive deficits after stroke [24,25,26,27]. However, only a few studies maintained methodological rigor in examining the optimal clinical cutoff for stroke patients. For example, education stratification in receiver operating characteristics (ROCs) was rarely applied [14]. This has a significant clinical impact, as many Asian studies report inadequate detection accuracy using the one additional point recommendation for the MoCA for patients with <12 years of education [28,29,30,31]. Furthermore, it is uncertain which cutoffs should be used in societies with greater educational disparities [32,33,34]. In brief, increasing evidence reveals that sociocultural considerations are indispensable in interpreting the results of the MMSE and MoCA.

The brief and broad nature of the MMSE and MoCA render them practical and popular in clinical settings, particularly in developing countries in Asia where resources are limited. It is commonplace that only patients showing prominent functional impairments are referred for further neuropsychological evaluation. However, such services are often inaccessible to underserved groups in the community (e.g., poor health, low income, rural areas). Thus, accurate detection for PSCI is crucial while patients are in the hospital. Is it possible to balance the limitations of the MMSE and MoCA with practicality for the benefit of both patients and clinicians? This question is worthy of exploration due to the 5–15% higher prevalence of dementia due to stroke in Asia than in North America and Europe [35]. Although cognitive screening is part of stroke care protocol, whether the MMSE and MoCA are clinically useful in Asia remains unclear.

The aim of this review is to compare the sensitivity and specificity of the MMSE and MoCA in Asia. Based on this, recommendations for future practice and research will be outlined. While there are other cognitive tests currently available—e.g., ACE-III (Addenbrooke’s Cognitive Examination 3rd Edition) [36] and the IQCODE (Informant Questionnaire for Cognitive Decline in the Elderly) [37]—this review focused on the MMSE and MoCA because (1) ACE-III was designed to differentiate AD and frontotemporal dementia, (2) the IQCODE is an informant-based structured questionnaire—as opposed to the MMSE and MoCA, which directly measure the patient’s cognitive function—and (3) the MMSE and MoCA remain the most well-known cognitive tests across multidisciplinary settings in Asia. Sensitivity and specificity were chosen as indices of detection accuracy because they are not dependent on the prevalence of PSCI in the population.

## 2. Methods

### 2.1. Brief Description of the MMSE and MoCA

The MMSE evaluates 6 cognitive domains, i.e., memory, orientation, registration, attention, language, and visuoconstruction ability. It has a maximum score of 30 and a recommended cutoff score of <24 for dementia [38]. Although it was originally sampled with a variety of dementing conditions—e.g., psychosis, affective disorders [3]—it was not designed for stroke, and has shown to be inadequate for PSCI [14,15]. It has also been criticized for its lack of executive tasks [4]. 

The MoCA addresses this limitation by adding executive tasks [4,19]. It also measures language, memory, attention, abstraction, and orientation, with a maximum score of 30. A cutoff score of <26 is recommended for MCI. Recent studies have challenged the clinical utility of the MMSE and MoCA for stroke patients [11,14,15,39].

### 2.2. Search Strategy

The PubMed, SCOPUS, Web of Science, and Science Direct databases were searched for relevant articles up to November 2020. Only full-text, peer-reviewed English articles were selected, using keywords containing “stroke” OR “cerebrovascular accident” AND “cognitive impairment” OR “cognitive deficits” AND “cognitive assessment” OR “screening” OR “test” OR “tool” AND “sensitivity” OR “specificity”. This review adhered to the Preferred Reporting Items for Systematic Reviews and Meta-Analyses (PRISMA) [40] guidelines; Figure 1 summarizes the process.

### 2.3. Eligibility Criteria

Cognitive impairments were operationalized as cognitive deficits measured by standardized neuropsychological battery/assessment or clinical rating scales. For the purpose of this review, we included studies that (1) recruited stroke patients aged 18 years old and above, (2) used the MMSE and/or MoCA as cognitive screening tools, (3) reported sensitivity, specificity, and area under the curve (AUC), and (4) involved subjects of Asian ethnic origin, residing within the Asian continent. Studies were excluded if they (1) recruited incompatible subjects, e.g., animals, or people with other neurological, neuropsychiatric, or medical conditions, (2) used neuroimaging, electroencephalography, or brain stimulation as their primary method, (3) were reviews, protocols, or opinion papers, or (4) used an incompatible study design, e.g., retrospective cohort.

### 2.4. Quality Assessment

Results were evaluated using the Quality Assessment of Diagnostic Accuracy Studies version 2 (QUADAS 2) [41], and judged according to the risk of bias and applicability of the selected studies. This consists of 4 domains: patient selection, index test, reference standard, and flow and timing. Two independent reviewers (J.K. and P.S.) assessed this information. Case–control design studies were included where controls were believed to be a representative sample of the population, i.e., cases and controls were enrolled from the same population pool.

### 2.5. Data Synthesis

Sample size, sensitivity and specificity for optimal cutoffs, and cognitive impairment incidents were extracted into Review Manager 5.4 (RevMan 5.4) [42] to calculate true positives (TP), false positives (FP), true negatives (TN), and false negatives (FN). Since over half of the studies were case–control studies, estimates of prevalence were not calculated.

## 3. Results

The search yielded 1306 records. After removing duplicates, 846 articles remained, and were screened by title and abstracts. Following this, 810 were excluded due to geographical locations, sample (e.g., dementia, Parkinson’s disease, brain injury), non-cognitive outcomes (e.g., functional ability), review papers, randomized controlled trials, or neuroimaging studies. This resulted in 48 articles for further full-text evaluation, of which 11 articles met the inclusion criteria and were included in this review; 7 of these studies were conducted in China, 3 in Singapore, and 1 in Korea. The National Institute of Neurological Disorders and Stroke–Canadian Stroke Network 5-Minute Protocol (NINDS-CSN 5) [43] was included as it consists of the original subtests in the MoCA, i.e., five-word memory task, six-item orientation, and one-letter phonemic fluency. Moreover, two of the three studies examining the NINDS-CNS 5-Minute reported ≥200 participants [44,45].

### 3.1. Risk of Bias

Six of the eleven included studies (54.5%) attributed an unclear risk of bias in patient selection to case–control designs. Nevertheless, there were five prospective cohort studies (41.7%), lending strength to the overall quality of current review. Risk of bias and applicability concerns are reported in Figure 2.

### 3.2. Sample Characteristics

The MMSE was reported in four studies with 901 subjects, the MoCA in nine studies with 2154 subjects, and the NINDS-CNS 5 in three studies with 680 subjects. This resulted in a total of 3735 participants. Eight studies reported more than 200 subjects (72.7%). 

### 3.3. Analysis

Forest plots were created in RevMan 5.4 using the following data: SE, SP, number of participants, and positive and negative incidents. Based on this, a summary ROC (SROC) was constructed to visually explore the diagnostic accuracy of index tests (see Figure 3 and Figure 4). Q index and bivariate model SROC were not examined due to the small number of studies [46]. Moreover, performing this analysis could be misleading for clinicians in other Asian countries because Mandarin was the dominant language in over 90% of the studies. 

### 3.4. Detection Accuracy of the MMSE and MoCA

Four studies that compared the MMSE and MoCA showed equivalent sensitivity and specificity to identify PSCI [47,48,49,50]. However, only two studies [47,49] met the detection accuracy standard of 80% sensitivity and 60% specificity, as suggested by Stolwyk et al. [14]. In the work of Dong et al. [48], there was a substantial difference in sensitivity between the optimal (MMSE = 61%, MoCA = 69%) and recommended (MMSE = 71%, MoCA = 78%) cutoff scores. Detection accuracy improved after a processing speed test was added in the ROC analysis (sensitivity = 97–98%, specificity = 76–78%). Likewise, Zhu et al. [50] reported poor sensitivity for both tests (MMSE = 68%, MoCA = 64%). Studies that found equivalent detection accuracy for the MMSE and MoCA also recruited older patients (see Table 1). Out of the nine studies that examined the MoCA, six reported adequate sensitivity (78–97%) and specificity (64–90%). Only one study showed poor specificity [51], attributable to the ceiling effect among patients with higher education levels. Two [45,52] of the three studies that examined the NINDS-CNS 5 demonstrated adequate sensitivity (82–92%) and specificity (67–68%). Another study showed fair SE (70%) but good SP (82%) [44].

Qualitatively, studies showing adequate sensitivity and specificity had several characteristics: (1) shorter time interval between stroke and screening (i.e., ≤4 days), (2) older patients, and (3) lower dropout rates in prospective cohort designs (<30%).

### 3.5. Covariates

Although the MMSE and MoCA both appear adequate for detecting PSCI, there are several covariates to consider.

#### 3.5.1. Education

Many studies (82%) reported significantly lower education in patients with PSCI. Despite the considerable education weightage on PSCI, only two studies stratified patients according to education in the ROC analysis [45,51]. The 12-year education cutoff for MoCA was found to be inadequate [53]; instead, a 6-year or primary education level cutoff better fit the Asian population [31,45,47,48,50,51]. Over half of the studies included in the review raised concerns over education’s effect on MoCA scores (see Table 2).

#### 3.5.2. Age

Eighty percent of the studies that reported inadequate sensitivity and specificity for the MMSE and MoCA recruited younger stroke patients (61–64 years old). In comparison, studies that reported adequate sensitivity and specificity recruited older patients (68–73 years old). Overall, 73% of studies showed that patients with poorer cognitive outcomes were significantly older.

#### 3.5.3. Stroke Characteristics

Seven studies (64%) recruited patients with mild stroke or TIA, and eight studies (73%) excluded patients with aphasia. Less than half of the studies reported stroke location [44,47,48,50,53], and only one study reported stroke lateralization [50]. Only two studies did not report stroke severity [31,51], i.e., using the NIHSS (National Institutes of Health Stroke Scale) [54]. The remaining studies reported mild severity of stroke.

#### 3.5.4. Time since Stroke

There was no clear pattern indicating that cutoff scores for the MMSE and NINDS-CNS 5 are affected by the time interval between screening and stroke. With regard to the MoCA, one study showed reduced sensitivity when cognitive screening was conducted 6 months post-stroke (sensitivity = 63%) [55]; nevertheless, it was suitable to identify moderate-to-severe PSCI [44]. In contrast, if screening was performed within 2 weeks after stroke, the MoCA showed adequate sensitivity [44,47,49,51,53], except in the study by Zhu et al. [50].

#### 3.5.5. Cognitive Domains

Only two studies stratified cutoff scores according to cognitive domains in the ROC analysis [48,51]. Visuospatial, executive function, abstraction, memory, and language tasks in the MoCA reached a ceiling effect for patients with >12 years of education [51]. Both screening tests further showed poor sensitivity in detecting single-domain memory deficits in stroke patients (MMSE = 67%, MoCA = 68%). The MMSE showed poorer sensitivity in detecting non-memory cognitive deficits compared to the MoCA [48]. No studies examined praxis, neglect, or number processing in their neuropsychological assessments.

Overall, preliminary results suggested adequate detection accuracy for the MMSE and MoCA. However, this must be considered carefully with respect to the critical covariates listed above.

## 4. Discussion

While the MMSE and MoCA are widely used in stroke settings in Asia, this is the first review to address concerns about the psychometric properties of both tools for Asian stroke patients. An optimal cutoff score has the best trade-off between SE (true positive) and SP (true negative). However, to pool together the sensitivity and specificity of the MMSE and MoCA for a single cutoff score for the Asian population would undermine the diversity of cultures, ethnicities, and languages; the paucity of high-quality studies in other parts of Asia further deters this.

Studies that directly compared the MMSE and MoCA found them to be equivalent in detecting PSCI, but at varying accuracy levels. In other words, despite their equivalence, some studies found both screening tests to be inadequate for stroke patients [48,50]. These studies reported large sample sizes (*N* = 229–400), over 50% dropout rate at follow-up, and younger patients. The extent to which these factors are statistically significant remained a question because of the limited number of studies (*N* = 4); nevertheless, high dropout rate can erroneously estimate PSCI, e.g., in aging studies, dropouts were prevalent among individuals with worse white matter integrity [55,56]. In a study by Dong et al. [48], sensitivity improved by approximately 20% after a visuomotor processing speed test was added in the ROC analysis. This aligns with recent studies suggesting visual processing speed as an underlying cognitive function that affects performance in other cognitive domains in neurocognitive disorders [57,58]. Further evidence is warranted in Asia to determine whether the addition of a visual processing speed task can improve the detection accuracy of the MMSE and MoCA in stroke settings. 

On the other hand, the MoCA generally showed adequate sensitivity and specificity for stroke patients. A closer examination of the findings supports the importance of education stratification in ROC analysis [59]. For example, many cognitive tasks in the MoCA (e.g., executive, memory, abstraction) were inadequate for stroke patients with higher education [51]. This can partly explain the poor specificity (47%) reported. It was postulated that some tasks in the MoCA are easy for patients with higher levels of education, risking an underestimation of PSCI. Arguably, a recent study in Israel found that the MoCA was difficult even for healthy and highly educated older adults [60]. In contrast, a floor effect was reported for stroke patients with lower education, suggesting that the test items in MoCA are too difficult for this group [50]. Similar findings have been reported in previous studies [7,22,61,62,63]. In this light, what may potentially contribute to the observed limitations? While sociocultural differences and stroke characteristics must be acknowledged, it is difficult to ignore the limitation of global cognitive screening tests—using a universal cutoff score to identify cognitive deficits. Recent evidence points towards domain-specific screening tests that minimize verbal requirements and emphasize clinical utility, e.g., informing clinicians of potential rehabilitative targets [11].

In this review, studies that found adequate sensitivity also reported older patient samples, concurrent with epidemiological studies showing poorer cognitive outcomes after stroke with increasing age [1,62,64]. It is plausible that the MMSE and MoCA appear capable of accurate detection when results merely reflect a sociodemographic artefact. However, the age factor can also be intertwined with education, e.g., it was not mandatory for older individuals to obtain a formal education in past decades in developing countries [65].

One way to accommodate the aforementioned challenges is to provide normative data stratified by age and education, but this is financially demanding and time-consuming for developing countries to achieve. It may be feasible to pool data across Asia to provide appropriate age- and education-based cutoff scores. It may also be relevant to supplement the MoCA with additional cognitive tests, e.g., for processing speed [48,66]. Nevertheless, concern arises regarding qualification and skills, as administering additional cognitive tests means enlisting the expertise of neuropsychologists. Misinterpretation of results and liability due to misreporting can prove to be counterproductive. A possible alternative is to minimize the use of single cutoff scores and shift towards domain-specific scores, as increasing evidence shows that this provides a more sensitive measure for PSCI [12,39,63]. Clinicians are recommended to adjust cutoff scores for the MoCA based on education level, i.e., an additional 1 point for <6 years. Previous studies in Singapore and China further support this recommendation [28,51]. More studies are warranted to determine whether these recommendations improve detection accuracy for PSCI. For instance, illiteracy can be a potential confounding factor in community settings [61]. 

Despite methodological limitations pertaining primarily to education, it may still be worthwhile to propose a screening test to guide Asian clinicians in stroke management. In general, the MoCA appears to be more robust than the MMSE for mild stroke patients with higher education levels. For those with lower education, it is postulated that both tests will likely show comparable detection accuracy for PSCI. However, clinicians should note that this does not necessarily indicate good sensitivity or specificity. The IQCODE could be a complementary assessment [67], albeit this requires a reliable informant who might not be perceptive of the subtle cognitive deficits seen in stroke patients. In addition, the NINDS-CNS 5 can discriminate between patients with and without PSCI beyond 3 months since stroke onset, despite having only three tasks, i.e., memory, orientation, fluency; it may also be suitable to conduct over the telephone [52], allowing clinicians flexibility during the COVID-19 pandemic; however, this would clearly exclude patients with language impairments. We recommend the NINDS-CNS 5 for individuals with high cerebrovascular risk factors [68] and stroke patients in settings where the MoCA is not feasible due to time or resource constraints. From a statistical viewpoint, the MoCA should be prioritized over the NINDS-CNS 5 because having a greater number of test items can reduce the probability of random errors. 

The MoCA is also valid in the acute stroke phase (≤14 days), and scores can predict cognitive deficits in mild stroke 3–6 months post-ictus. Beyond the acute phase, it is not plausible to postulate whether the MoCA will remain valid. Attention should be given to cognitive domains with poorer scores to guide rehabilitative goals. A pass/fail global score in the MMSE and MoCA is reductionist, and gives little insight into rehabilitation targets. Studies have demonstrated that cognitive screening tests designed specifically for stroke, with domain-specific results, provide more clinically meaningful information, e.g., the Oxford Cognitive Screen (OCS) [11,12,39]. The OCS is also freely available in Mandarin [69], Cantonese [70], and Malay [71]. This test includes tasks that measures visual attention (neglect), praxis, and executive function, commonly observed to be impaired among stroke patients. Further evidence is warranted to determine the efficacy of the OCS in Asia. 

For future clinical research, several important considerations are outlined. Firstly, stroke lateralization and location should be reported where possible, as this affects the presentation of cognitive deficits. For example, memory impairments are prominent in posterior cerebral artery infarcts [66,72]. Next, the time between screening and stroke should be examined, as early testing can be impractical, wasteful of limited resources, and distressing for patients and their caregivers if results are inaccurate [73]. Spontaneous neural recovery in the first few months of stroke may also influence detection accuracy [22,74]. Moreover, bilingualism is characteristic of former Western colonies—e.g., Singapore, Malaysia, Hong Kong, and India—and is traditionally associated with social and economic advantages. Studies have shown higher cognitive reserves in bilinguals and poorer verbal but better executive abilities [75,76,77,78,79]. The detection accuracy of the MMSE and MoCA for bilingual stroke patients in Asia remains unknown. Finally, future research should strive for prospective cohort designs, as case–control designs can artificially inflate estimation of PSCI. It is important to have healthy controls as a comparison group due to high vascular risk factors among stroke patients, which can contribute to cognitive decline [80]. 

This review highlights the importance of education stratification in influencing the detection accuracy of the MMSE and MoCA, but there are some limitations: (1) the majority of the included studies focused on mild ischemic stroke and excluded aphasic patients, creating selection bias where some patients could not participate due to more severe disability; (2) 7 of the 11 studies were conducted in China, limiting the generalizability of findings in other Asian countries; and (3) over half of the studies were case–control designs, potentially introducing bias in sampling and, thus, inflating detection accuracy. However, the total sample size obtained in this review was relatively large (*N* = 3735), and most studies used neuroimaging data for evidence of stroke rather than self-report/hospital admission notes. Furthermore, all of the studies reported region-specific cutoff values using the ROC analysis. Five studies were prospective cohort designs, balancing the limitations of case–control designs. This review also provides support for a 6-year education cutoff for the MoCA and recommendations for clinical practice in Asia. Although positive prediction value and negative prediction value were not discussed in this review, we wish to highlight that they provide greater clinical utility [81,82]. However, as they are dependent on prevalence, we believe that it is currently not possible to compare between studies. These data were included at the discretion of clinicians where the prevalence of PSCI was known (see Appendix A).

## 5. Conclusions

Although the MMSE and MoCA are routinely used in clinical settings in Asia, only a limited number of studies examined their sensitivity and specificity for PSCI. While both tests generally showed adequate detection accuracy, many studies were plagued by the lack of educational stratification in determining cutoff scores and exclusion of patients with aphasia. Considering their invariable influence on accurate detection, clinicians are advised to repeat cognitive screening within the first few months of stroke. It is beneficial for future studies to investigate whether domain-specific cognitive screening can ameliorate this limitation. This review calls for further research in developing nations in Asia.

## Figures and Tables

**Figure 1 ijerph-18-08962-f001:**
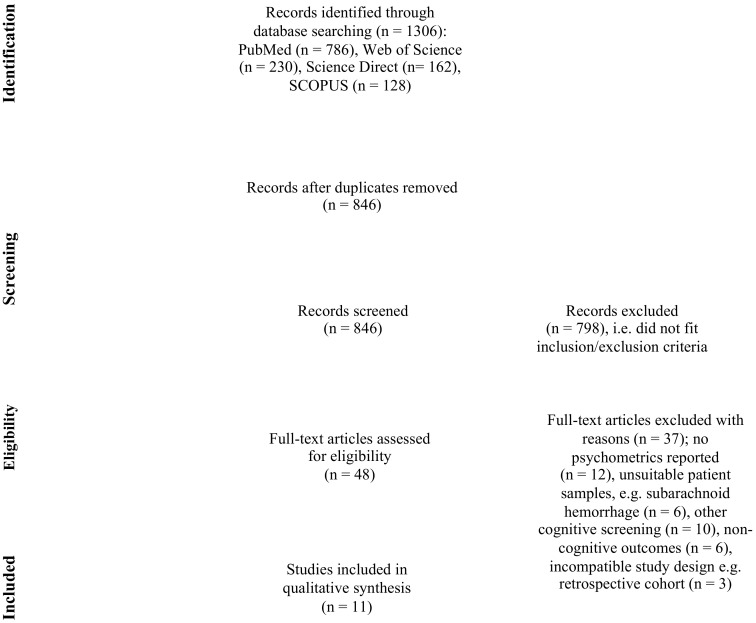
PRISMA (Preferred Reporting Items for Systematic Reviews and Meta-Analyses) flowchart for screening tests for post-stroke cognitive impairments in Asia.

**Figure 2 ijerph-18-08962-f002:**
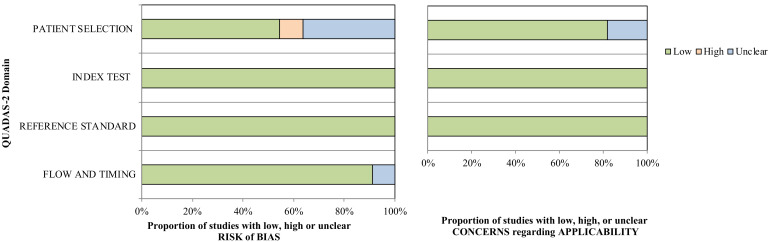
QUADAS-2 risk of bias and concerns of applicability chart; QUADAS-2: Quality Assessment of Diagnostic Accuracy Studies version 2.

**Figure 3 ijerph-18-08962-f003:**
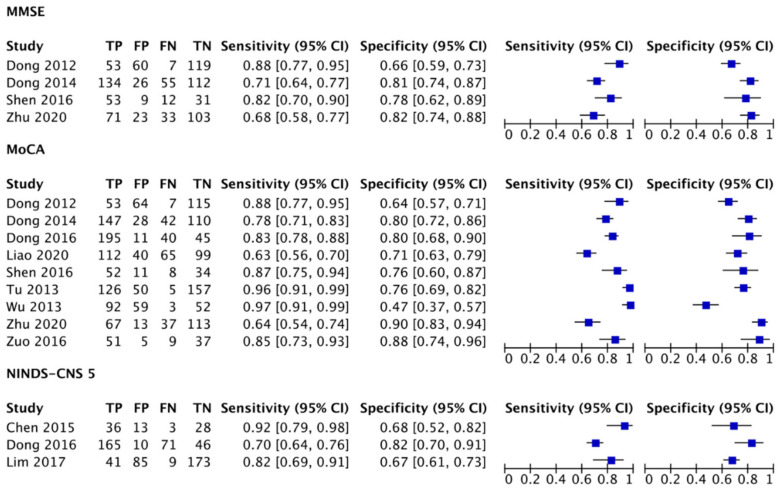
Forest plots depicting the SE and SP of the MMSE, MoCA, and NINDS-CNS 5; MMSE: Mini-Mental State Examination; MoCA: Montreal Cognitive Assessment; NINDS-CNS 5: The National Institute of Neurological Disorders and Stroke–Canadian Stroke Network 5-Minute Protocol; TP: true positive; FP: false positive; FN: false negative; TN: true negative; CI: confidence interval.

**Figure 4 ijerph-18-08962-f004:**
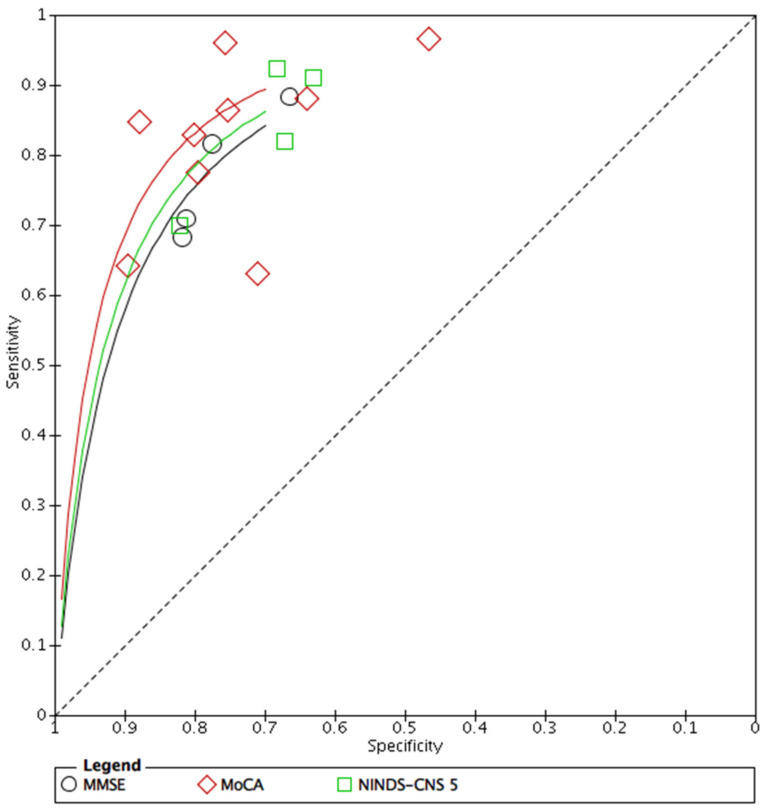
Exploratory SROC curve for diagnostic accuracy of the MMSE, MoCA, and NINDS-CNS 5; MMSE: Mini-Mental State Examination; MoCA: Montreal Cognitive Assessment; NINDS-CNS 5: The National Institute of Neurological Disorders and Stroke–Canadian Stroke Network 5-Minute Protocol; SROC: summary receiving operating characteristic.

**Table 1 ijerph-18-08962-t001:** Summary of studies and covariates categorized by tests.

MMSE													
Study	Disease	Language	Study Design	Sample Size	Age	Education	NIHSS	Time Since Stroke	Cutoffs	Sensitivity (95% CI)	Specificity (95% CI)	PPV	NPV
												(%)	(%)
Dong 2012	VCI	Preference	Prospective	239	67.9 *	5.7 *	1 to 4	3 days	≤25/26	0.88 (0.77, 0.95)	0.66 (0.59, 0.73)	47	94
Dong 2014	VCI	Preference	Prospective	400	64.3 *	6.3 *	0 to 1	2.4–3.4 months	≤26	0.71 (0.64, 0.77)	0.81 (0.74, 0.87)	84	67
Shen 2016	VCI-ND	Mandarin	Case–control	104	70.6 *	8.7	3.16	≤14 days	≤27/28	0.82 (0.70, 0.90)	0.78 (0.62, 0.89)	82	78
Zhu 2020	VCI	Mandarin	Prospective	229	63.8	6*	1	≤14 days	≤27	0.68 (0.58, 0.77)	0.82 (0.74, 0.88)	71	23
**MoCA**													
Dong 2012	VCI	Preference	Prospective	239	67.9 *	5.7 *	1 to 4	3 days	≤21/22	0.88 (0.77, 0.95)	0.64 (0.57, 0.71)	45	94
Tu 2013	VCI-ND, VD	Changsha	Case–control	470	69.4–73.2	6.3–8.1 *	?	≥3 months?	≤26/27	0.96 (0.91, 0.99)	0.76 (0.69, 0.82)	86	93
Wu 2013	VCI-ND	Mandarin	Case-control	206	68.1	8.65	?	Acute?	22/23	0.97 (0.91, 0.99)	0.47 (0.37, 0.57)	N/A	N/A
Dong 2014	VCI	Preference	Prospective	400	64.3 *	6.3 *	0 to 1	2.4–3.4 months	≤23	0.78 (0.71, 0.83)	0.80 (0.72, 0.86)	84	72
Dong 2016	VCI-ND	Preference	Prospective	291	68.4 *	5.5*	1 to 4	2.6–4 days	≤20/21	0.83 (0.78, 0.88)	0.80 (0.68, 0.90)	50	95
Shen 2016	VCI-ND	Mandarin	Case–control	104	70.6 *	8.7	3.16	≤14 days	≤23/24	0.87 (0.75, 0.94)	0.76 (0.60, 0.87)	86	75
Zuo 2016	VCI	Mandarin	Case–control	102	58.3 *	Level *	1	10 days	≤22/23	0.85 (0.73, 0.93)	0.88 (0.74, 0.96)	91	80
Liao 2021	VCI	Mandarin	Case–control	316	61.1 *	Level *	2	6 months	≤24	0.63 (0.56, 0.70)	0.71 (0.63, 0.79)	74	60
Zhu 2020	VCI	Changsha	Prospective	229	63.8	6 *	1	≤14 days	≤21	0.64 (0.54, 0.74)	0.90 (0.83, 0.94)	N/A	N/A
**NINDS-**													
**CNS 5**													
Chen 2015	VCI	Mandarin	Case–control	80	62.9	7.2 *	2	10 months	24	0.92 (0.79, 0.98)	0.68 (0.52, 0.82)	73	90
Dong 2016	VCI-ND	Preference	Prospective	291	68.4 *	5.5 *	1 to 4	2.6–4 days	≤7/8	0.70 (0.64, 0.76)	0.82 (0.70, 0.91)	49	92
Lim 2017	VD	Korean	Prospective	308	69.1 *	Level	5	3 months	≤6/7	0.82 (0.69, 0.91)	0.67 (0.61, 0.73)	33	95
Wei 2020	VCI	Mandarin	Case–control	2989	63	Level	1.16	1–2 months	≤10	0.91 (0.89,0.92)	0.63 (0.60, 0.65)	71	87

* Significant differences between comparison groups; VCI: vascular cognitive impairment; VCI-ND: vascular cognitive impairment, no dementia; VD: vascular dementia; CI: confidence interval; PPV: positive predictive value; NPV: negative predictive value; MMSE: Mini-Mental State Examination; MoCA: Montreal Cognitive Assessment; NINDS-CNS 5: The National Institute of Neurological Disorders and Stroke–Canadian Stroke Network 5-Minute Protocol; NIHSS: National Institutes of Health Stroke Scale.

**Table 2 ijerph-18-08962-t002:** Methods for education adjustment across studies.

MMSE				
Study	Adjusted	Additional One Point	Method	Notes
Dong 2012	Yes	< primary level education	Regression	Cutoff scores did not differ between patients with lower (≤6 years) and higher educational levels
Dong 2014	Yes	< primary level education	ROC analysis	Cited lack of education stratification for cutoff as study limitation
Shen 2016	No	<12 years		Due to small sample size
Zhu 2020	Yes	<6 years	ROC analysis	
**MoCA**				
Dong 2012	Yes	< primary level education	ROC analysis	Cutoff scores did not differ between patients with lower (≤6 years) and higher educational levels
Tu 2013	No			Regression analysis showed education’s effect
Wu 2013	Yes	<12 years	Cutoff scores stratified by education level	Not education-adjusted MoCA ≤ 22/23 Education ≤ 6 years MoCA ≤ 15 Education 6–12 years MoCA ≤ 22 Education > 12 years MoCA ≤ 23
Dong 2014	Yes	< primary level education	ROC analysis	Cited lack of education stratification for cutoff as study limitation
Dong 2016	Yes	< primary level education	ROC analysis	Education-adjustment did not affect cutoff scores
Shen 2016	No	<12 years		Cutoff scores not adjusted for education
Zuo 2016	No	<12 years		Authors recommended education-adjusted cutoff scores for future studies
Liao 2020	No	<12 years		Authors recommended education-adjusted cutoff scores for future studies
Zhu 2020	Yes	<6 years	ROC analysis	MoCA is more suitable for educated individuals
**NINDS-CNS 5**				
Chen 2015	Yes	Not applicable	Analysis of variance	Cutoff scores not adjusted for education
Dong 2016	Yes	Not applicable	ROC analysis	Education-adjustment did not affect cutoff scores
Lim 2017	Yes	Not applicable	Logistic regression	Categorized education as ≤6 years vs. >6 years

MMSE: Mini-Mental State Examination; MoCA: Montreal Cognitive Assessment; NINDS-CNS 5: The National Institute of Neurological Disorders and Stroke–Canadian Stroke Network 5-Minute Protocol; ROC: receiving operating characteristic.

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
