# Peer review of "Current Update on the Clinical Utility of MMSE and MoCA for Stroke Patients in Asia: A Systematic Review"

_ijerph, 2021, doi:10.3390/ijerph18178962_

Round 1
Reviewer 1 Report
This paper presents a review which explore the clinical utility of two tests, the MMSE and MoCA, for stroke patients – and particularly for post-stroke dementia assessment - in Asia and provide recommendations for clinical practice.
The paper is well written, clear and well supported. Interest is strong and should be of interest to the clinical community in Asia. Overall, I would recommend the authors to take into account the following comments to improve the quality and the robustness of the manuscript.
Major comments
Comment 1. Post-stroke dementia
The authors must define the “post-stroke dementia”, and add the acronym (PSD), the first time they mentioned it, as well as the acronym (PSCI) for the term: “post-stroke cognitive impairment”.
Moreover, I suggest the authors include the following reference:
Mijajlović et al. (2017). Post-stroke dementia – a comprehensive review. BMC Medicine. DOI 10.1186/s12916-017-0779-7
Comment 2. The MMSE and MoCA
The authors said that the MMSE and the MoCA tests “were originally designed to screen for cognitive impairments related to Alzheimer’s disease”.
Regarding the MMSE, the authors (Folstein, 1975) said that the “MMS was given to 69 patients chosen specifically as clear examples of clinical conditions: 29 with dementia syndromes due to a variety of brain diseases, 10 with affective disorder, depressed type with clinically recognizable cognitive impairment, and 30 with uncomplicated affective disorder, depressed type”. When given to these 69 patients, “the test proved to be valid and reliable and it was able to separate the three diagnostic groups”.
Folstein, S. Folstein, P.R. McHugh, “Mini-mental state”: A practical method for grading the cognitive state of patients for the clinician, Journal of Psychiatric Research, Volume 12, Issue 3, 1975, https://doi.org/10.1016/0022-3956(75)90026-6.
Regarding, the MoCA test, the objective was “to develop a 10-minute cognitive screening tool to assist first-line physicians in detection of mild cognitive impairment (MCI), a clinical state that often progresses to dementia” (Nasreddine, 2005).
Nasreddine ZS, Philllips NA, Bédirian Vet al. The Montreal Cognitive Assessment (MoCA): a brief screening tool for mild cognitive impairment. J Am Geriatr Soc (2005); 53:695–699.
Thus, the authors should precise the original design and construct of the tests.
Comment 3. The Informant Questionnaire on Cognitive Decline in the Elderly
The authors focused their review on the MMSE and MoCA tests, and said: “The MMSE and MoCA are the most used screening tools for cognitive impairments after stroke (Quinn 38 et al., 2018).”
Although this choice is legitimate, the authors should explain why they did not retain the IQCODE test. Indeed, “another approach in screening PSD is using informants’ reports on changes in the patients’ functional status, an example of which is the Informant Questionnaire on Cognitive Decline in the Elderly” (Tang WK, et al., 2003). These authors said that the IQCODE is the most commonly employed assessment, but also said: “When used as a sole instrument, IQCODE does not appear to be useful in screening PSDE in Chinese elderly”.
Tang WK, et al. Can IQCODE detect poststroke dementia? Int J Geriatr Psychiatry. 2003;18:706–10.
Comment 4. Results
The method applied to retain the 48 articles for further full-text evaluation is unclear. Indeed, the search yielded 1,306 records, and firstly 846 articles remained (removing duplicates) and were screened by title and abstracts. Secondly, 810 were excluded due geographical locations, sample. Perhaps the two processes were not carried out in a serial way but on the same basic dataset (the 1306 records).
Comment 5. Discussion
The authors point out the differences within Asian countries and suggest that demographics, such as bilingualism, social and economic advantages, should be taken into account.
In this vein, the authors should take into account, a large national Korean study of vascular cognitive impairment (VCI) epidemiology, using the Korean version of the VCIHS neuropsychological protocol in a multicenter, hospital-based stroke cohort in Korea. The study was applied on a large sample size (620 subjects with ischemic stroke within 7 days of symptom) and defined PSD as any major cognitive impairment seen at more than 3 months after a stroke event, regardless of pre-stroke cognitive status (Yu, 2013).
Yu KH et al. Korean-Vascular Cognitive Impairment Harmonization Standards Study Group. Cognitive impairment evaluated with vascular cognitive impairment harmonization standards in a multicenter prospective stroke cohort in Korea. Stroke. 2013;44:786–8.
Minor comments
The English in the present manuscript require minor improvement. Please carefully proof-read spell check to eliminate grammatical errors.
The image quality of the figure 3 must be improved, in particularly for the text, on left.
Author Response
Dear Reviewer,
Please find attached our revised manuscript, entitled “Current update on the clinical utility of MMSE and MoCA for stroke patients in Asia: A systematic review”.
We gratefully thank the Reviewer for his/her valuable comments and
suggestions. We have strictly followed the Reviewer’s comments. We managed to address all the comments and suggestions point by point in the separate document as attached. The details of the revisions to the manuscript were indicated using ‘track changes’ function. In fact, we made English editing of the whole manuscript and better arranged the whole manuscript as suggested by reviewer.
We hope that the manuscript is now ready for publication.
Best regards,
Ponnusamy Subramaniam, on behalf of all co-authors

Reviewer 2 Report
To. authors
The prevalence of dementia, including cognitive impairment, increases after stroke, which is a distinct trend in Asian countries. Therefore, MMSE and MoCA may be useful for conveniently screening cognitive impairment in stroke patients. However, it is questionable whether these screening tests are useful in assessing cognitive impairment in stroke patients. This is because most Asian studies overlooked the characteristics and educational level of stroke patients, and cutoff values are not well established in these patients. In this context, this review investigated the specificity and sensitivity of MMSE, MoCA, and NINDS-CNS 5 in Asian stroke patients using meta-analysis, and provided advice for future studies. This is a well-written study which deals with important data. Here are my suggestions:
Major points:
1. In Figure 1, the arrow is missing. Also, please indicate the number of articles per database and excluded patients per exclusion criteria.
2. Wasn't retrospective study excluded from the analysis? Authors need to more clearly state the inclusion/exclusion criteria in the Eligibility criteria.
3. Did the studies not consider the stroke subtype? Is only mild stroke and TIA defined based on initial stroke severity included? It would be desirable to include the stroke subtype/initial stroke severity score in the table.
4. Authors need to provide a brief overview of MMSE, MoCA, and NINDS-CNS 5 in Method.
5. Please indicate in the table the studies that have corrected the academic level.
6. In Results and Discussion, the contents of NINDS-CNS 5 are missing.
7. When comparing MMSE, MoCA and NINDS-CNS5, which test do the authors think is the best in stroke patients? Are they all similar? By any chance, does the recommended test change depending on the acute and chronic periods? It would be great if these were explained in Discussion.
Minor points:
1. Information about abbreviations and symbols is missing from the captions of all tables and figures.
2. There is a minor grammatical error.
3. Does the abbreviation SE and SP mean sensitivity and specificity respectively? Abbreviation information is missing from the manuscript.
4. What is cutoff 24 of NINDS-CNS 5 in Table 1? Isn't the total score 12 points?
Author Response
Dear Reviewer,
Please find attached our revised manuscript, entitled “Current update on the clinical utility of MMSE and MoCA for stroke patients in Asia: A systematic review”.
We gratefully thank the Reviewer for his/her valuable comments and
suggestions. We have strictly followed the Reviewer’s comments. We managed to address all the comments and suggestions point by point in the separate document as attached. The details of the revisions to the manuscript were indicated using ‘track changes’ function. In fact, we made English editing of the whole manuscript and better arranged the manuscript as suggested by referees.
We hope that the manuscript is now ready for publication.
Best regards,
Ponnusamy Subramaniam, on behalf of all co-authors

Round 2
Reviewer 1 Report
The authors have addressed most of my concerns from my previous review.
The manuscript was really improved.
The authors should replace: “cognitive presentations of Alzheimer’s disease” with “cognitive presentation of Alzheimer’s disease”